# Synchronized Efficacy and Mechanism of Alkaline Fertilizer and Biocontrol Fungi for *Fusarium*
*oxysporum* f. sp. *cubense* Tropical Race 4

**DOI:** 10.3390/jof8030261

**Published:** 2022-03-03

**Authors:** Yuanqiong Li, Shuting Jiang, Jiaquan Jiang, Chengxiang Gao, Xiuxiu Qi, Lidan Zhang, Shaolong Sun, Yinhai Dai, Xiaolin Fan

**Affiliations:** 1College of Natural Resources and Environment, South China Agricultural University, Guangzhou 510642, China; 18843185299@163.com (Y.L.); jiangshuting075@163.com (S.J.); jqjiang163163@163.com (J.J.); chengxianggao@163.com (C.G.); qixiuxiu0318@163.com (X.Q.); lidanzhang@scau.edu.cn (L.Z.); sunshaolong328@scau.edu.cn (S.S.); Sigri259427@163.com (Y.D.); 2R&D Center of Environment Friendly Fertilizer Science and Technology, South China Agricultural University, Guangzhou 510642, China

**Keywords:** alkaline fertilizer, antioxidation system, banana Fusarium wilt, biocontrol fungi, root activity, synergistic effect, tylosis

## Abstract

The purpose of this study was to determine the effect and mechanism of alkaline fertilizer, bio-control fungi, and their synergistic application on control of Fusarium Tr4 incidence. Synchronized use of the alkaline fertilizer and biocontrol fungi eliminates rhizome browning and reduces the incidence rate of banana Fusarium wilt. The incidence of yellow leaves (ratio of yellow leaf to total leaf) and disease index in +Foc Tr4 CF treatment were the same (65%), while incidence of yellow leaves and disease index in +Foc Tr4 AFBCF were 31% and 33%, respectively. Under the stress of Foc Tr4 infection, the synergistic utilization of the alkaline fertilizer and biocontrol fungi would raise the activities of peroxidase, catalase and superoxide dismutase in banana roots. The root activity of banana was also increased. As a result, the banana height and stem diameter increments, shoot and root dry weight, accumulation of N, P and K in banana plants had been increased. The efficacy of the synergistic application of alkaline fertilizer and biocontrol fungi was not only reducing Foc Tr4 pathogen colonization and distribution in banana plants, but also preventing tylosis formation in vascular vessel effectively. Therefore, the normal transport of water and nutrients between underground and aboveground is ensured.

## 1. Introduction

According to FAO statistics, banana (*Musa* spp. AAA) is of great importance to the world food security as a fruit and food crop [1]. However, Fusarium wilt has threatened banana production and caused huge economic losses for a long time [2]. The banana Fusarium wilt, also known as Panama disease or banana wilt, is a soil-borne vascular bundle disease caused by *Fusarium oxysporum* f. sp. *cubense* Tropical Race 4 (Foc Tr4) [3]. It is highly contagious, destructive, and difficult to be effectively controlled [4,5]. Thus, one of the primary issues is to control Fusarium wilt effectively in banana cultivation.

The most widely used methods to control banana Fusarium wilt are chemical and agricultural measures as well as disease resistant variety release. The chemical control includes quaternary ammonium compounds, soil disinfectants, propiconazole and the other chemical agents which are used to kill pathogens in the soil directly [6,7,8]. The advantages of chemical control are quick, economic, simple and effective in a short period. However, soil physi-chemical properties and microorganism diversity will be destroyed by long-term application of the chemicals [9]. There also exist hidden environmental and food safety risks [10]. Therefore, breeding and releasing disease-resistant varieties are regarded as the most effective and safe measures to control the disease. A variety of banana wilt resistant new varieties have been bred through hybrid, mutation and transgenic breeding, such as Giant Cavendish Tissue Culture Variants (GCTCVs) [11]. However, breeding the new variety faces challenges such as a long cycle and unstable resistance. Additionally, promotion of the new variety has been constrained by long vegetation period and less fruit flavor compared to Cavendish banana. Therefore, no commercial resistant banana variety has been released to control Foc Tr4 disease [12]. The agricultural control measures are mainly crop rotation, soil improvement, water and fertilizer management and other tillage practices. Zhang et al. found that extract and volatile substances of Chinese leek could inhibit the occurrence of banana wilt disease. Banana and leek rotation could effectively control banana wilt [13]. Combination of lime with chemical fertilizers showed a positive effect in reducing incidence of banana wilt [10,14]. Although conventional agricultural measures could alleviate the occurrence of the banana wilt, the methods consume time and labor. Therefore, the agricultural measure is not suitable for modern intensive orchard to control the banana wilt. The difficulty of controling banana wilt is as follows. (1) The wilt is soil-borne fungi with a long survival in the soil (more than 20 years), even in the absence of plant hosts [15], or within alternate hosts which do not necessarily show disease symptoms [16]. (2) As a vascular pathogen, it escapes the contact with the control means (e.g., non-systemic fungicides, non-endophytic biological control agents, etc.) once it penetrates into the plant. (3) It can be spread by banana vegetative propagation material, soil vectored by workers and machinery and irrigation water, etc. [17]. (4) Monoculture of bananas, especially Cavendish varieties, facilitates the spread of Foc Tr4 pathogens [10]. (5) Soil acidity exacerbates banana wilt epidemic [18]. Single conventional agricultural measure is thus difficult to play with its effect. To control banana Fusarium wilt requires the cooperation of conventional agricultural measures and the other methods to exert their synergistic effects. 

Soil acidification which is caused by long-term excessive and irrational use of chemical fertilizer, especially nitrogen fertilizer in banana plantation results in the proliferation and spread of banana wilt pathogens [10,19,20]. Increasing soil pH will be a possible measure to decrease the incidence of banana wilt. According to our previous study [21,22], instead of conventional fertilizer by alkaline fertilizer could significantly increase soil pH and then reduce the occurrence of banana wilt. Although some studies have also shown that biocontrol has an obvious effect on banana Fusarium wilt [10,23,24,25], the efficacy will be poor if soil acidity is not improved [21,22]. Biocontrol is one of the hot topics of banana Fusarium wilt control before release of the disease resistant variety. It is found that many antagonistic microorganisms can be used for biocontrol with good results [26,27,28]. Among these antagonistic microorganisms, numerous biocontrol fungi have been used to control the Foc Tr4. Results showed that pathogen prevention and control efficacy of multiple fungi was superior to that of a single one [29,30]. Our previous research found that synergism of *non-pathogenic Fusarium oxysporum*, *Trichoderma* and *Paecilomyces lilacinus* could significantly reduce the occurrence of banana Fusarium wilt. The results imply that the combination of the three fungi has a synergistic effect [31]. However, the majority of research on banana wilt control focus on single measure in general, which is difficult to achieve the desired efficacy. Previous studies found that the combination of alkaline fertilizer and actinomycetes could optimize the rhizosphere microbial structure, improve soil enzyme activity and increase banana disease resistance [21,22]. It can be concluded that the combination of various measures can give full play to their efficacy to control banana wilt and promote banana growth. However, the influence of BCF combined with alkaline fertilizer on the efficacy of banana wilt control is unknown. This paper intended to take combination use of the alkaline fertilizer with BCF as a control measure. The effect and mechanism of alkaline fertilizer and BCF on banana Fusarium wilt control were studied through physiological, biochemical, nutritional characteristics. The effect of the alkaline fertilizer and BCF on expansion and colonization of the *Fusarium oxysporum* f. sp. *cubense* Tropical Race 4 (Foc Tr4) in banana plants were studied by green fluorescent protein (GFP)-tagged Foc Tr4. The findings would be expected to provide a theoretical foundation for the coordinated control of banana wilt with alkaline fertilizer and BCF.

## 2. Materials and Methods

### 2.1. Materials

Cavendish banana cultivar Brazilian (*Musa* spp. AAA) was used in this study. Banana seedlings with 5 green leaves and average height of 5–6 cm was used to carry out sterilized sand culture experiment. Average plant height and stem diameter of plantlets were surveyed before transplanting. 

The biocontrol fungi (BCF) applied in the trail was composed of *non-pathogenic Fusarium oxysporum*, *Paecilomyces* Sp. and *Trichoderma harzianum* QL18-8 suspension with the equal number of spores according to a volume ratio of 1:1:1. The *Fusarium* strains were GFP-tagged *Fusarium oxysporum* f. sp. *cubense* Tropical Race 4 (Foc Tr4) and provided by the South China Agricultural University fungal laboratory. 

The fertilizer was liquid fertilizer, which included conventional fertilizer and alkaline fertilizer. The pH of the liquid fertilizers was 5.5–6.0 and 7.5–8.0 for conventional fertilizer and alkaline fertilizer, respectively. The fertilizer was prepared by adding urea (N content accounted for 46%), potassium chloride (K_2_O content accounted for 60%) and sodium dihydrogen phosphate (P_2_O_5_ content accounted for 45%) into urea-formaldehyde solution during the urea-formaldehyde formation. N:P_2_O_5_:K_2_O ratio of the liquid fertilizer was 2:0.5:1. 

The substrate was sea sand with a particle size between 0.5 mm and 3 mm. The sand was washed with tap water until salt was free. The sand was then autoclave sterilized twice after it was leached by deionized water. The sand was dried for subsequent use 450 g of sand was weighed and placed in a plastic pot of 12 cm in diameter and 10 cm in height. One plantlet was transplanted in each pot. Another 350 g was used to cover banana plantlets after transplanting. The pot experiment was conducted in the greenhouse of South China Agricultural University. 

### 2.2. Experimental Design

The experiment was in split-split plot design with pathogen in the main plots and BCF and fertilizer treatments in the sub-plots by use of Design-Expert 13. The primary factor included inoculation pathogen (+Foc Tr4) and non-inoculation pathogen (−Foc Tr4). The BCF included added and without added BCF. The fertilizer included conventional fertilizer and alkaline fertilizer. The eight treatments were defined as −Tr4 CF, −Tr4 AF, −Tr4 CFBCF, −Tr4 AFBCF, +Tr4 CF, +Tr4 AF, +Tr4 CFBCF, +Tr4 AFBCF, respectively. Thirty replicates were designed for each treatment, and one pot was one replicate. All treatments received the same amount of N, P and K. The N applied to each treatment was 0.2 g per kilogram sand. The fertilizer was dissolved in water and divided into 10 times to be applied once a week. The pH of the fertilizer solution was adjusted to 5.5–6.0 and 7.5–8.0 for conventional fertilizer and alkaline fertilizer before application, respectively. 

### 2.3. Methods

#### 2.3.1. Preparation of BCF

The *non-pathogenic Fusarium oxysporum*, *Trichoderma (Trichoderma harzianum* QL18-8) and *Paecilomyces lilacinus* were inoculated in Potato Dextrose Agar (PDA) medium, respectively. The fungus was activated under 26 °C condition for 7 days. Twenty fungi plugs of 8 mm in diameter were placed in 500 mL Potato Dextrose Water (PD water), respectively. After shaking 7 days at 180 rpm under 26 ± 0.5 °C, the fungus suspension was then filtered through eight layers of gauze to separate the hyphae. Then, the filtrate of non-pathogenic *Fusarium oxysporum*, *Trichoderma* and *Paecilomyces lilacinus* were diluted with sterile deionized water until the number of spores of each fungus was 3 × 10^7^ conidia/mL, respectively. Then the three diluted filtrates were mixed in a volume ratio of 1:1:1 to form the BCF.

#### 2.3.2. Preparation of Pathogen

Pathogen of Foc Tr4 was inoculated in PDA solid medium and activated under 26 °C for 7 days. Twenty fungi plugs of 8 mm in diameter were placed in 500 mL PD water. After shaking 7 days at 180 rpm under 26 ± 0.5 °C, the fungus suspension was then filtered through eight layers of gauze to separate the hyphae. Then, the filtrate was diluted with sterile deionized water until the number of spores was 10^5^ conidia/mL, which was the Foc Tr4 pathogen for inoculation.

#### 2.3.3. Banana Transplantation and Plant Inoculation

Healthy banana plantlets with similar height, diameter of pseudostem and same number of green leaves were selected. Roots of banana plantlets were immersed in 1000 times diluted chlorothalonil solution for a few seconds before transplanting. The banana plantlets were then inoculated with the BCF four weeks after transplantation by use of injured root inoculation method as follows. Four 5 cm deep holes were made in four directions of banana plantlets by use of sterilized glass rod. Thirty milliliter the BCF solution was poured into the four holes evenly, and the holes were covered with sand. The mock-inoculated control plants were treated with 30 mL sterilized water to each pot. Ten days later 30 mL of GFP-tagged Foc Tr4 spore suspension at 10^5^ conidia/mL was inoculated as above. The mock-inoculated control plants were also treated with 30 mL sterilized water to each pot.

### 2.4. Sample Collection and Determination

#### 2.4.1. Incidence of Yellow Leaves (IYL) and Disease Index (DI)

After inoculation of Foc Tr4, disease infection state of the banana was recorded by observation of leaf color change every 10 days and then the IYL and DI was calculated. Banana Fusarium wilt was divided into 6 levels [21]. Level 0 was healthy banana without symptom. Level 1 was banana suffered disease symptom less than 20%. Level 2 and 3 was the plant suffered disease symptom within 20–40% and 40–80%, respectively. Level 4 was banana suffered serious disease symptoms and only the top 1 to 2 leaves were healthy. Level 5 was the plant which had died. IYL and DI were calculated as follows.
$$\mathrm{IYL}\;\left(\%\right)=\frac{\;\mathrm{number}\;\mathrm{of}\;\mathrm{yellow}\;\mathrm{leaves}}{\mathrm{total}\;\mathrm{plant}\;\mathrm{leaves}}\times 100$$
$$\mathrm{DI}\;\left(\%\right)=\sum \frac{\left(\mathrm{number}\;\mathrm{of}\;\mathrm{infected}\;\mathrm{plants}\;\mathrm{at}\;\mathrm{each}\;\mathrm{level}\times \mathrm{value}\;\mathrm{at}\;\mathrm{this}\;\mathrm{level}\right)}{\mathrm{number}\;\mathrm{of}\;\mathrm{total}\;\mathrm{plants}\;\times \mathrm{highest}\;\mathrm{level}\;\mathrm{value}}\times 100$$

#### 2.4.2. Determining Browning Degree of Banana Rhizome

Three banana plants were randomly selected from each treatment 60 days after inoculation. Rhizome and pseudostem of the banana were cut into two parts longitudinally. Image of the rhizome and pseudostem section was taken by WinRhizer Sysytem. The browning area of rhizome was calculated by Image-Pro (MEDIA CYBERNETICS).

#### 2.4.3. Banana Biomass Determination

Five plants were randomly selected from each treatment to measure banana height and diameter of pseudostem 60 days after inoculation, and increment of the height and pseudostem diameter accounted. Banana roots, pseudostem and leaves were collected after the banana plant was poured out and sand attached to the root washed. Fresh weight of the roots, stems and leaves was measured, respectively. Samples were then headed at 105 °C for 30 min and dried at 75 °C to constant weight to account dry matter. The plant samples were stored for NPK test.
$$\mathrm{Height}\;\mathrm{increment}\;\left(\mathrm{cm}\right)=\mathrm{average}\;\mathrm{height}\;-\;\mathrm{average}\;\mathrm{height}\;\mathrm{before}\;\mathrm{transplanting}$$
$$\mathrm{Diameter}\;\mathrm{increment}\;\left(\mathrm{mm}\right)=\mathrm{average}\;\mathrm{diameter}\;-\;\mathrm{average}\;\mathrm{diameter}\;\mathrm{before}\;\mathrm{transplanting}\;$$

#### 2.4.4. Determination of Banana Root Activity and Root Antioxidant Enzymes

Three banana plants were randomly selected from each treatment 60 days after inoculation, and roots were collected as above. Root activity was measured by the triphenyl tetrazolium chloride method (TTC method) according to Liu et al. [32]. The antioxidant system of banana was evaluated by the activity of peroxidase (POD), catalase (CAT) and superoxide dismutase (SOD) of the roots. The activities of POD, CAT, and SOD were analyzed using ultraviolet-visible spectrophotometer method according to Lusso et al. [33], Tománková et al. [34] and Giannopolitis et al. [35].

#### 2.4.5. N, P and K Content Determination

The above dried roots, stems and leaves powder sample (passed to 0.25 mm sieve) was digested by hydrogen peroxide (H_2_O_2_)-sulfuric acid (H_2_SO_4_) [36]. N, P and K concentration of the digestion were measured by continuous flow analyzer (CFA, AMD Paris, France), SmartChem automated discrete analyzer (ADA) and digita flame analyer (DFA), respectively. The accumulative uptake of N, P and K was calculated, respectively.

#### 2.4.6. Microscopic Observation

The root 1 cm from rhizome, middle section of rhizome and three sections of pseudostem (from rhizome to the first leaf, from the first to the second leaf and from the second to the third leaf counting backward) of the plants inoculated with Foc Tr4 were examined for infection process and colonization at early stage of banana wilt disease. For microscopic observation, banana tissues were washed in sterile distilled water, then cut into thin sections along the vertical and longitudinal axes by hand sectioning. Then, the tissue sections were placed on a microscope slide, submerged in a water droplet, and covered with a glass cover slip. Microscopic observation was carried out under a Double-Scanning Laser Confocal Microscope (NIKON A1) equipped with filter blocks with spectral properties matching the fluorescence of GFP by taken Honghong Dong et al. as reference [37]. Three plants were prepared for each sample, and each experiment was repeated thrice.

#### 2.4.7. SEM Image Observation

The SEM samples were prepared according to Nebesářová’s method [38]. Root samples 1 cm from rhizome were collected in +Tr4 CF and +Tr4 AFBCF treatments 60 days after inoculation. The samples were cut into segments of 5 mm and fixed in 2.5% glutaraldehyde for 24 h. The fixed specimens were rinsed twice in phosphate buffer solution for 30 min per time. Then the samples were dehydrated by 30%, 50%, 60%, 70%, 80%, 90% and 100% ethanol for 15 min step by step, respectively. The samples were washed twice with tertiary butyl alcohol for 30 min each step. The washed samples were critical-point-dried by Freeze dryer. The dry samples were sprayed with gold by Polaron sputter coater and observed by CARL ZEISS EVO 10 to take SEM image. 

### 2.5. Data Analysis

Results statistics were completed with Design-Expert 13. The data in tables and figures are mean ± SE, where SE is the standard error. The difference among treatments was estimated by Univariate ANOVA and multiple comparison.

## 3. Results

### 3.1. Synergistic Effect of Alkaline Fertilizer and BCF on Banana Fusarium Wilt

Bananas inoculated with Foc Tr4 showed obvious symptoms of wilt disease two months later, while bananas without Foc Tr4 inoculation grew healthily. When banana plants suffered banana wilt, their old leaves started yellowing and wilting and the symptoms gradually spread to new leaves. For severe symptoms, new leaves turned yellow and died. The number of banana leaves of +Tr4 CF treatment was the least, and all the leaves except the top one were withered and yellow. The bananas of +Tr4 AFBCF treatment showed less Fusarium wilt symptoms. Only old leaves were yellowing in comparison with those of Foc Tr4 inoculation treatment (Figure 1).

Figure 2 showed that colors of longitudinal section of rhizome and pseudostem were normal, which means no infection of Foc Tr4 in −Tr4 treatment. However, the color of the section changed into brown, and symptoms were significantly different among four fertilizer and biocontrol treatments in +Tr4 one. The brown-colored area expanded from inside to outside and gradually to pseudostem 60 days after Foc Tr4 inoculation. The dark brown color means the banana had been infected by Foc Tr4, the tissue had been damaged and even the xylem blocked by tylosis. The darker of the color the bigger of the area, the severer the disease. The brown area of rhizome was in order of +Tr4 CF > +Tr4 AF > +Tr4 CFBCF > +Tr4 AFBCF. The largest brown area, accounted for 77.9% of the whole rhizome section area, was found in the +Tr4 CF treatment (Table 1). However, the areas of CFBCF and AFBCF treatment were smaller than those of CF and AF one, respectively. The brown rear in AFBCF rhizome was the least, accounting for only 15.5% of the whole area. The results showed that combination use of the BCF and alkaline fertilizer could be capable of banana healthy growth. The synergistic effect of the alkaline fertilizer and BCF on banana Fusarium wilt control was also shown by reducing brown area of pseudostem or prevent xylem blockage (Figure 2).

### 3.2. Synergistic Effect of Alkaline Fertilizer and BCF on IYL and DI

The incidence of yellow leaves (IYL) and disease index (DI) of banana were significantly decreased by the combination use of alkaline fertilizer and BCF. As shown in Figure 3, the IYL and DI in Foc Tr4 treatments increased with time. Symptoms of Fusarium wilt were observed about 20 days after the Foc Tr4 inoculation. The IYL and DI curve of +Tr4 AFBCF and +Tr4 CFBCF showed an exponential increase. The IYL and DI curve was linearly increased in the first 20 days after the symptom appeared, and then the curves entered decreasing growth stage. The IYL and DI curve of +Tr4 AF showed a linear increase during the whole experiment period. However, the IYL and DI curve of +Tr4 CF treatment was sigmoid type. The results indicated that there existed two inflection points over time. In the first 10 days after the Fusarium wilt onset, IYL and DI increased linearly. They were raised rapidly in 30~40 days after inoculation and then decreased gradually after 40 days. During the whole experiment period, the IYL and DI of +Tr4 CF were the highest, while the IYL and DI of +Tr4 AFBCF was the lowest among the four +Foc Tr4 treatments. It was confirmed that application of the alkaline fertilizer and BCF simultaneously could effectively control the banana Fusarium wilt. However, IYL curves of four treatments in −Foc Tr4 were almost coincide. The average slope (0.2864) of the four curves was much lower than that (0.8477) of the four in +Foc Tr4. The yellow leaves in −Foc Tr4 treatment were only 15.7–19.3% and they were the normal senescence of old leaves as the plant growth. Nevertheless, the IYL was 31.2%, 40.7%, 56.7% and 64.9% for +Tr4 AFBCF, +Tr4 CFBCF, +Tr4 AF and +Tr4 CF, respectively (Figure 3a). The DI of the four treatments in −Foc Tr4 were almost zero (Figure 3b), while the DI of +Tr4 AFBCF, +Tr4 CFBCF, +Tr4 AF and +Tr4 CF were 33.3%, 40.7%, 60.5% and 65.4% (Figure 3b), respectively. There was a significantly synergistic effect of alkaline fertilizer and BCF on reducing IYL and DI.

### 3.3. Synergistic Effect of Alkaline Fertilizer and BCF on Banana Growth

Banana normal growth was affected by inoculation of the Foc Tr4 pathogen. Compared with banana without inoculation, the height and stem diameter increments, shoot and root dry weight of −Foc Tr4 were decreased by 25.5%, 20.3%, 13.4%, and 28.6%, respectively (Table 2, Average 2, *p* = 0, 0.057, 0.041, 0.05, respectively). No matter how the banana was inoculated Foc Tr4 or not, the height and stem diameter increments, shoot and root dry weight were increased when the BCF applied with fertilizer together (Table 2, Average 1). Among CF, AF, CFBCF, AFBCF treatments with and without Foc Tr4 inoculation, the banana height and stem diameter increments, shoot and root dry weight of the AFBCF treatment were the largest one. Banana growth was dramatically limited after the plant was infected by Foc Tr4 pathogen. However, even when the banana was inoculated Foc Tr4 pathogen, application of the BCF combined with fertilizer, especially alkaline fertilizer, was the best measure to promote banana growth. Compared with CF, plant height and stem diameter increments, shoot and root dry weight of CFBCF were increased by 17.8%, 79.7%, 37.8% and 47.8%, respectively (*p* < 0.05). The corresponding values of AFBCF were 47.9%, 158.5%, 59.0% and 50.7%, respectively (*p* < 0.05). The results showed that the synergistic effect of the BCF and alkaline fertilizer on banana growth was more significant than that of single BCF or its combination with conventional fertilizer. In the four treatments of −Foc Tr4, the banana height and stem diameter increments, shoot and root dry weight of AFBCF treatment were increased by 30.4%, 61.2%, 64.2% and 93.1%, respectively, compared with those of CF (*p* < 0.05, Table 2). In conclusion, the application of alkaline fertilizer in combination with the BCF had significant synergistic effects on banana resistance to Fusarium wilt. 

### 3.4. Synergistic Effect of Alkaline Fertilizer and BCF on Banana Root Activity

Banana root activity was affected by inoculation of the Foc Tr4 pathogen. As shown in Figure 4, the root activity of banana inoculated with Foc Tr4 was lower than that of without Foc Tr4 inoculation (Figure 4a). On average, the root activity of +Foc Tr4 treatments was 59.6% lower than that of −Foc Tr4 treatments (*p* < 0.01, Figure 4b). Regardless of whether the pathogen was inoculated or not, the combination of fertilizer and BCF resulted in higher root activity than fertilizer alone. (Figure 4a). On average, the root activity in treatments of fertilizer and BCF was increased by 120.8% compared with that of fertilizer treatment alone in −Foc Tr4. The corresponding value was 45.0% in +Foc Tr4 (*p* < 0.05, Figure 4c). The results proved that root activity would be decreased when the banana plant suffered from Foc Tr4 pathogen infection. However, the root activity could be raised by the synergistic influence of fertilizer and BCF. The root activity of CFBCF and AFBCF treatments was increased by 26.8% and 81.5% (*p* < 0.05, Figure 4a) compared with CF treatment in +Foc Tr4, respectively. The corresponding values were 165.9% (CFBCF) and 179.3% (AFBCF) (*p* < 0.05) in −Foc Tr4, respectively. The results indicated that the synergistic effect of fertilizer and BCF on banana root activity was greater when there was no pathogen infection. In summary, the root activity of banana plant would be decreased when the banana was infected by the Foc Tr4 pathogen. The synergism of the fertilizer and BCF could increase the root activity, and the synergistic application of alkaline fertilizer and BCF was superior to that of conventional fertilizer.

### 3.5. Synergistic Effect of Alkaline Fertilizer and BCF on Antioxidant Enzyme Activities of Banana Root

Table 3 showed the effects of the alkaline fertilizer and BCF on the enzyme activity of banana root. The activity of POD, CAT and SOD in banana root were increased by 46.44%, 59.49% and 23.94% in +Foc Tr4 treatment as compared with −Foc Tr4 (Table 3, average 2, *p* < 0.01), respectively. In other words, after banana was threatened by the Foc Tr4 pathogen, the antioxidant enzyme activity could be increased. The application of fertilizer and BCF together would result in increase in the POD and SOD activity of banana root, regardless of whether the pathogen was inoculated or not (Table 3, average 1). The POD and SOD activity of banana root treated with the BCF was increased by 3.42% to 24.26%, respectively, compared with that without BCF application. Nevertheless, CAT activity was reduced by 19.16% for the treatment −Foc Tr4. Among the +Foc Tr4 treatments, POD, CAT and SOD activity were increased by 13.46%, 23.26% and 4.40%, respectively, when treated with BCF (*p* < 0.05). Thus, the antioxidant enzyme activity of banana root was increased after the plant was stressed by the Foc Tr4 pathogen. The synergistic effect of the fertilizer (including alkaline fertilizer and conventional fertilizer) and BCF on the enzymes was more significant. The POD, CAT and SOD activity increment of banana root was from 4.18% to 36.62% in +Foc Tr4 treatments. The POD, CAT and SOD activity of AFBCF treatment were the biggest one among the four treatments when the banana suffered with Foc Tr4 stress (Table 3). In conclusion, the greater synergistic efficacy to prevent the banana wilt disease is cooperation use of alkaline fertilizer and BCF together when the banana plant faces the Foc Tr4 threat. 

### 3.6. Synergistic Effect of Alkaline Fertilizer and BCF on NPK Nutrients Absorption and Accumulation of Banana

Nitrogen (N), phosphorus (P) and potassium (K) nutrition state reflects the effect of alkaline fertilizer and BCF on disease resistance and banana growth. As shown in Figure 5, each treatment had a significant influence on N absorption and accumulation in banana (Figure 5a). The N accumulation in +Foc Tr4 treatments was decreased by 20.7% compared to that in −Foc Tr4 (Figure 5b, *p* = 0.005). The N absorption and accumulation by banana could be improved by combination use of fertilizer and the BCF in case with or without Foc Tr4 inoculation. Without Foc Tr4 inoculation, the N accumulation of the combination use of fertilizer and the BCF was increased by 39.0% compared to that of BCF and fertilizer alone. However, with the Foc Tr4 inoculation, the corresponding value was 32.7% (Figure 5c). The N adsorption and accumulation would be reduced after banana was infected by the Foc Tr4 pathogen, while the N uptake could be improved through application of fertilizer and the BCF together. The synergistic effect of alkaline fertilizer and BCF was more significant than that of conventional fertilizer and BCF (Figure 5a). Without Foc Tr4 inoculation, the N accumulation of CFBCF and AFBCF was increased by 27.3% and 80.0% compared with that of CF (*p* < 0.05), respectively. However, with Foc Tr4 inoculation, the N accumulation of CFBCF and AFBCF was increased by 48.4% and 74.9% (*p* < 0.05), respectively. The synergistic application of alkaline fertilizer and BCF was greater due to their combination could resist the damage of Foc Tr4 to banana growth. More N absorbed and accumulated by banana plant would promote its health growth and to resist the disease in turn. The results showed that the synergistic effect of alkaline fertilizer and BCF on eliminating banana Foc Tr4 infection was that the combination of alkaline fertilizer and BCF improved plant nitrogen nutrition and promoted banana health growth to resist the disease.

The effect of pathogens, fertilizers and BCF on P accumulation in banana was similar as on N (Figure 6a). P accumulation was reduced by 18.1% on average in +Foc Tr4 treatments (Figure 6b, *p* = 0.059). Utilization of BCF and fertilize could significantly increase the P accumulation no matter how the pathogen was inoculated or not. The P accumulation was increased by 54.4% in FBCF treatment in −Foc Tr4 treatment. However, it increased by 44.8% in +Foc Tr4 one (Figure 6c). Thus, fertilizer combined with BCF could significantly improve phosphorus uptake and accumulation in banana plant when it was infected by Foc Tr4 pathogen. The synergistic effect of alkaline fertilizer and BCF on P accumulation was greater than that of conventional fertilizer and BCF (Figure 6a). P accumulation of CFBCF and AFBCF was increased by 41.0% and 111.0%, respectively, compared with that of CF when the banana was not inoculated Foc Tr4. The corresponding values were 61.6% and 89.9%, respectively, when the banana was inoculated Foc Tr4. The reason why application fertilizer and BCF together resulted in increase in P accumulation might lie in their synergy could prevent banana Fusarium invasion. The great amount of P accumulated in the banana would also provide more P nutrients and promote plant healthy growth in turn. In conclusion, although P absorption and accumulation by banana could be reduced when the plant suffered banana Foc Tr4 pathogen infection, application fertilizer and the BCF together could significantly promote the P accumulation to ensure banana healthy growth to resist the disease.

As shown in Figure 7a, the K accumulation in banana plant was remarkably influenced by pathogens inoculation, fertilization and the BCF application. The K accumulation decreased by 21.7% in +Foc Tr4 treatment (Figure 7b, *p* = 0.014). The fertilizer combined with BCF could improve the K accumulation in banana no matter how the pathogen was inoculated or not. In −Foc Tr4 treatment, the K accumulation was increased by 55.9% with the application of the BCF. However, in +Foc Tr4 treatment, the K accumulation was raised by 40.7% in FBCF (Figure 7c). Although pathogen infection would reduce the K accumulation in banana, fertilizer combined with BCF was able to increase the K absorption and accumulation. The synergistic effect of alkaline fertilizer and BCF on the K accumulation was greater than that of conventional fertilizer and BCF (Figure 7a). The K accumulation of CFBCF and AFBCF increased by 31.4% and 105.6%, respectively, compared with that of CF in −Foc Tr4. However, the corresponding values were 65.4% and 95.6%, compared with that of CF in +Foc Tr4. The synergistic application fertilizer and BCF together could result in K accumulation and prevent banana Fusarium invasion. The great amount of K accumulated in the banana would improve plant K nutrition and be beneficial for banana healthy growth in turn. 

In conclusion, the results indicated that the synergistic application of fertilizer and BCF was a successful strategy to increase N, P and K nutrients of banana under Fusarium wilt stress. The nutrient accumulation was in order of K > N > P. The synergistic efficacy of the alkaline fertilizer and BCF was greater than that of the conventional fertilizer and BCF. In banana cultivation, it is suggested that alkaline fertilizer with high K, medium N and low P formula combined with BCF should be the priority for banana fusarium wilt control.

### 3.7. Mechanism of Synergistic Effect of Alkaline Fertilizer and the BCF on Banana Wilt Control

To study the effect of fertilizer and BCF on expansion and colonization of pathogens in banana plants, the status of *F. oxysporum* f. sp. *cubense* in the roots, rhizomes and pseudostems were examined at early stage of banana suffered wilt disease. This result indicates that Foc Tr4 hyphae can expand from the infected roots to the rhizomes and then pseudostems. The synergistic effect of alkaline fertilizer and the BCF were not only increase in banana root activity (Figure 4), antioxidant enzyme activity (Table 3) and NPK accumulation (Figure 5, Figure 6 and Figure 7), its combination with BCF could restrain the expansion and colonization of the Foc Tr4 pathogen in banana plant (Figure 8). The present microscope observation images showed that although the Foc Tr4 could be observed in banana roots, rhizomes and pseudostems, the conidia and hyphae of Foc Tr4 attached to the three organs was in the order of +Tr4 CF > +Tr4 AF > +Tr4 CFBCF > +Tr4 AFBCF (Figure 8). The expansion and colonization of the Foc Tr4 pathogens were the least in banana plant treated with alkaline fertilizer and the BCF (Figure 8 +Foc Tr4 CFBCF and +Foc Tr4 CFBAF). The Foc Tr4 pathogen was not only easily infected the roots of conventional fertilizer banana, but the pathogen easily penetrated and expanded from root upward to rhizome and pseudostem as well (Figure 8 +Foc Tr4 CF and +Foc Tr4 AF). Only a little amount of conidia and hyphae of Foc Tr4 was observed in root in +Tr4 AFBCF. There was no expansion of the pathogen from root to the other organs in the +Tr4 AFBCF treated banana plant. The results showed that in the alkaline fertilizer and BCF (Figure 8 +Foc Tr4 AFBCF) treatment there were no conidia and hyphae of the Foc Tr4 observed in rhizomes, pseudostem 1 and pseudostem 2, whereas abundance hyphae was detected in +Tr4 CF treated banana. The synergistic efficacy of alkaline fertilizer and BCF on the banana wilt control might lie in the following two aspects. Firstly, the Foc Tr4 pathogen infection and colonization in the root could be reduced significantly. Secondly, application the alkaline fertilizer and the BCF together could effectively stop the Foc Tr4 pathogen expansion speed from the root to the rhizome and subsequently to the pseudostem. Therefore, the possible reason of synergistic effect of alkaline fertilizer and the BCF on Fusarium wilt control was to prevent the infection and reduce the Foc Tr4 penetration and expansion from banana root upward to rhizome and pseudostem. Thus, there would be no tylosis formation in vascular vessels and no vessels blockage.

As shown in Figure 9, the electron microscope image of the banana root vascular vessels after it was infected with the Foc Tr4 pathogen. After inoculation with the Foc Tr4, the root vessels in +Tr4 CF treatment were almost completely blocked by the tylosis (Figure 9a). However, most of the vessels in +Tr4 AFBCF treatment were normal (Figure 9b). In terms of the internal structure of a single vessel, the tylosis in the +Tr4 CF treated vessel showed overlapping structures, and the vessel was completely blocked (Figure 9c). As a result, the transport of water and nutrients between the underground and aboveground parts of the banana was interrupted, eventually leading to death. However, little tylosis was found in vessels of the +Tr4 AFBCF treatment (Figure 9d). Thus, water and nutrients transport between the underground and aboveground parts would not be influenced and bananas grew normally under the treatment. This result was consistent with the finding microscope observation (Figure 8). In conclusion, the synergistic efficacy of the alkaline fertilizer and BCF treatment was not only reducing Foc Tr4 pathogen colonization and distribution in banana plants, but also preventing tylosis formation in vascular vessel effectively. Therefore, the normal transport of water and nutrients between underground and aboveground is ensured.

## 4. Discussion

It is well-known that banana Fusarium wilt is a destructive disease in the world. Soil acidification will aggravate the occurrence and prevalence of the Fusarium wilt [10]. Replacing conventional fertilizer by alkaline fertilizer could not only provide nutrients, but also was able to adjust soil pH to be suitable for banana and unsuitable for the pathogen growth [21,22]. Biocontrol is a hot topic of banana Fusarium wilt control. Although biocontrol is recognized an efficient measure to against banana wilt, acid soil will significantly reduce the efficacy of biocontrol in the field. This paper combined the application of the alkaline fertilizer with biocontrol fungi (BCF), which consist of *non-pathogenic Fusarium oxysporum*, *Trichoderma*, and *Paecilomyces lilacinus*, as banana wilt control measure. The results showed that the combination of alkaline fertilizer with the BCF could significantly reduce IYL, rhizome browning degree, and banana Fusarium wilt incidence. The reason why BCF could control banana wilt disease was that the BCF was inoculated before the Foc Tr4 pathogen. In other words, the *non-pathogenic Fusarium oxysporum* in BCF firstly occupied the same site in banana body that Foc Tr4 would infect. Thus, infection chance by the Foc Tr4 pathogen would be reduced. Other studies had also shown that *non-pathogenic Fusarium oxysporum* could compete for infection sites in root and could trigger plant defense reactions and induce systemic resistance [39]. In addition, the Trichoderma and Paecilomyces lilacinus in the BCF could further resist the invasion of banana wilt pathogens [28,40]. The mechanism to promote banana growth by use of the BCF and alkaline fertilizer may be as follows. Firstly, the *non-pathogenic Fusarium oxysporum* preferentially occupied the infection site of pathogenic *Fusarium oxysporum* and reduced the infection of Foc Tr4. Secondly, the synergistic effect of the three fungi in the BCF increased the resistance to the invasion of Foc Tr4 pathogen. Thirdly, alkaline fertilizer could create a suitable pH, which was not conducive to the germination and infection of the Foc Tr4 spores. Therefore, the combination of the BCF with alkaline fertilizer was necessary to give full play to the BCF to control banana Fusarium wilt.

Root activity will reflect the quality and metabolic status of roots to a certain extent; therefore, the root activity can be used to evaluate the growth status of plant. The stronger the root activity, the better the banana’s ability to absorb nutrients [32]. The root activity was significantly increased by synergistic effects of alkaline fertilizer and biocontrol fungi. The absorption and accumulation of nitrogen, phosphorus and potassium nutrients in banana plants were then promoted. Meanwhile, the banana growth was also improved by the alkaline fertilizer combined with BCF. The results were similar to previous studies on the effects of alkaline fertilizer and its combination with biological control agent on banana plant growth [21,22]. After banana plants were infected with Foc Tr4, the physiological mechanism of the plant against the disease was to produce secretions to defend the pathogen from expanding upward along the vessels [41]. Unfortunately, these secretions were accumulated inside of the vessels to form tylosis [5]. While preventing the spread of pathogens in the vessels, the tylosis blocked the vessels and affected the transport of water and mineral nutrients, ultimately leading to banana wilt and death [42]. The pathogen in the infected banana could also produce Fusarium acid to harm plants [43,44]. Additionally, pathogen would alter the synthesis of carbohydrates in plants, then impairing plant growth [45]. The reason why banana growth was promoted by application of the alkaline fertilizer and BCF was that their combination use could prevent the invasion and colonization of the pathogens in the plant. In addition, it was reported that the plant growth could be promoted, and pathogen invasion was constrained by the three fungi in the BCF [46,47,48].

Under stress conditions, plants will regulate their own antioxidant systems to maintain normal growth. Superoxide dismutase (SOD), peroxidase (POD) and catalase (CAT) are recognized to be the main enzymes of the plant antioxidant system. Cheng et al. (2020) and Dvorak et al. (2021) found that the reactive oxygen metabolic balance was able to be maintained by resisting and eliminating reactive oxygen and oxygen free radicals through increase in POD, CAT and SOD activities in plant when the plant was infected Foc Tr4 pathogen. Thus, the damage of reactive oxygen species to membrane was prevented and integrity and stability of the membrane system were maintained. Finally, pathogen invasion and reproduction were stopped, while plant resistance was enhanced [49,50]. Lin et al. (2021) found that the defense system of host plant could be activated by biological control agents and the ability to resist pathogen invasion was improved [51]. The present study showed that the antioxidant enzyme activity of banana roots could be improved after banana infection with Foc Tr4. The antioxidant enzyme activity in treatment applied alkaline fertilizer and BCF was higher than that of the others. Thus, the synergistic effect of the alkaline fertilizer and BCF was that reactive oxygen free radicals were removed by the cooperation of the BCF and alkaline fertilizer, and their damage to the structure and cell membranes were reduced. As the results, the resistance of bananas to Foc Tr4 pathogens was increased, and the incidence of Fusarium wilt was decreased.

Banana wilt would be observed when the root of the plant was infected by the pathogen and the pathogen expanded to pseudostems [52]. The similar results that Foc Tr4 could penetrate the vascular bundle tissues of the roots and expand to rhizomes and pseudostems fastly were reported in the literature [37,53,54]. If pathogen colonized in roots would be avoided or the expansion speed could be confined in banana roots, the banana Fusarium wilt control could be achieved. This study found that combining alkaline fertilizer and BCF would significantly reduce Foc Tr4 pathogen infection and colonization in the root. Moreover, application the alkaline fertilizer and the BCF together could effectively stop the Foc Tr4 pathogen expansion speed from the root to the rhizome and subsequently to the pseudostem. Therefore, the possible reason of synergistic effect of alkaline fertilizer and the BCF on Fusarium wilt control was to prevent the infection and reduce the Foc Tr4 penetration and expansion from banana root upward to rhizome and pseudostem. After banana was infected with Fusarium wilt, the plant would produce tylosis to block the vessel and prevent the pathogen expanding [55]. However, when the vessel is blocked, the transport of water and nutrients is interrupted. The banana growth was then impacted greatly [43]. The use of alkaline fertilizer in combination with BCF would reduce the formation of tylosis and result in less vessel blockage. Therefore, water and nutrients transport between the underground and aboveground parts would not be influenced and bananas grew normally.

## 5. Conclusions

(1)The combination application of alkaline fertilizer and biocontrol fungus reduces incidence of yellow leaves, rhizome browning degree, and banana Fusarium wilt incidence significantly.(2)The synergistic effects of alkaline fertilizer and biocontrol fungus increase root activity and root antioxidant enzyme activity of banana plants remarkably. Meanwhile, the absorption and accumulation of nitrogen, phosphorus and potassium nutrients in banana plants are boosted, so as to improve the growth of banana and enhance the plant to resist the Fusarium wilt disease.(3)Synchronized application of alkaline fertilizer and biocontrol fungi reduces the Foc Tr4 pathogen infection and colonization in banana root and spreading to rhizome and pseudostem. Tylosis formation is then prevented and thus the vascular vessel blockage is avoided to ensure the normal transport of water and nutrients between underground parts to aboveground ones.

## Figures and Tables

**Figure 1 jof-08-00261-f001:**
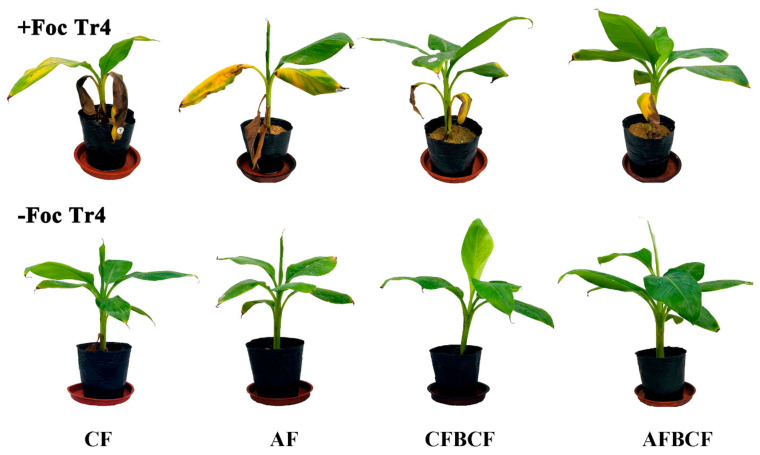
Comparison of banana growth state 60 days after Foc Tr4 inoculation. CF stands for conventional fertilizer. AF stands for alkaline fertilizer. CFBCF stands for conventional fertilizer and biocontrol fungi. AFBCF stands for alkaline fertilizer and biocontrol fungi.

**Figure 2 jof-08-00261-f002:**
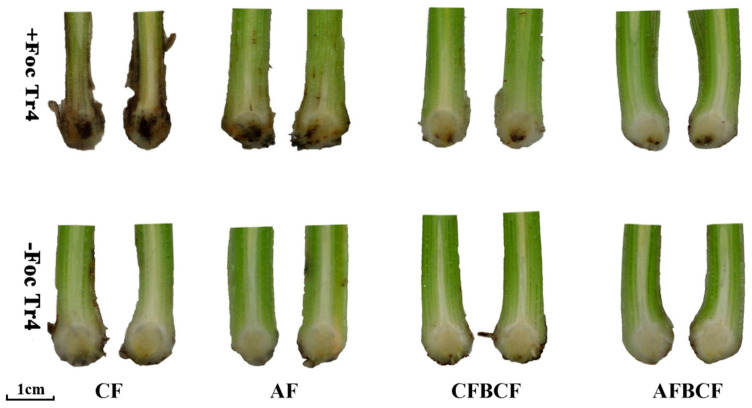
Comparison of browning degree of banana rhizome 60 days after Foc Tr4 inoculation. CF stands for conventional fertilizer. AF stands for alkaline fertilizer. CFBCF stands for conventional fertilizer and biocontrol fungi. AFBCF stands for alkaline fertilizer and biocontrol fungi.

**Figure 3 jof-08-00261-f003:**
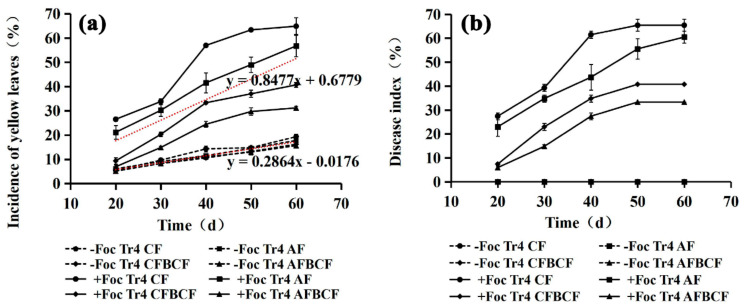
The effect of alkaline fertilizer and biocontrol fungi on incidence of yellow leaves (IYL) and disease index (DI) of banana plants. (**a**) Incidence of yellow leaves for each treatment; (**b**) Disease index for each treatment. The two equations are trend-line model of average value of IYL of four +Foc Tr4 treatments (**top**) and the that of four −Foc Tr4 treatments (**bottom**), respectively. CF stands for conventional fertilizer. AF stands for alkaline fertilizer. CFBCF stands for conventional fertilizer and biocontrol fungi. AFBCF stands for alkaline fertilizer and biocontrol fungi.

**Figure 4 jof-08-00261-f004:**
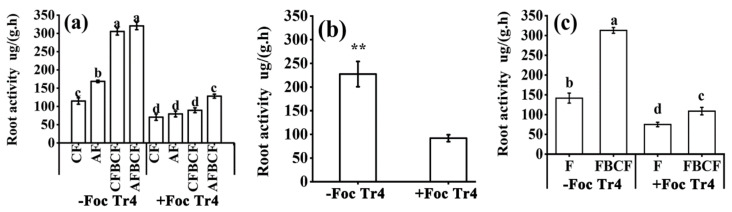
Effects of alkaline fertilizer and biocontrol fungi on banana root activity. (**a**) Effect of each treatment on root activity. (**b**) Effect of inoculation of pathogen (main plots) on banana root activity. (**c**) Effect of biocontrol fungi (sub-plots) on banana root activity. Different small case letters on the bars indicate significant difference (*p* < 0.05) among the treatments. ** indicated that there was extremely significant difference between inoculation and without inoculation of the Foc Tr4 (*p* ≤ 0.01). CF stands for conventional fertilizer. AF stands for alkaline fertilizer. CFBCF stands for conventional fertilizer and biocontrol fungi. AFBCF stands for alkaline fertilizer and biocontrol fungi.

**Figure 5 jof-08-00261-f005:**
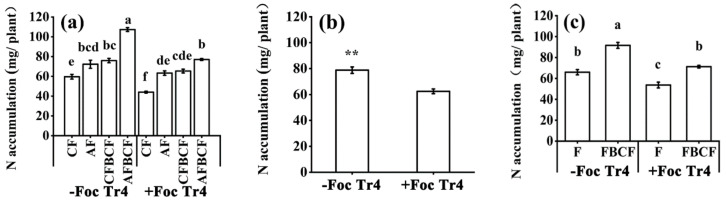
Effects of alkaline fertilizer and biocontrol fungi on N accumulation of banana 60 days after Foc Tr4 inoculation. (**a**) Effect of each treatment on N accumulation. (**b**) Effect of inoculation of pathogen (main plots) on N accumulation. (**c**) Effect of biocontrol fungi (sub-plots) on N accumulation. Different letters on pot of the bars indicated significant difference (*p* < 0.05) among the treatments. ** indicated that there was extremely significant difference between −Foc Tr4 and +Foc Tr4 (*p* < 0.01). CF stands for conventional fertilizer. AF stands for alkaline fertilizer. CFBCF stands for conventional fertilizer and biocontrol fungi. AFBCF stands for alkaline fertilizer and biocontrol fungi.

**Figure 6 jof-08-00261-f006:**
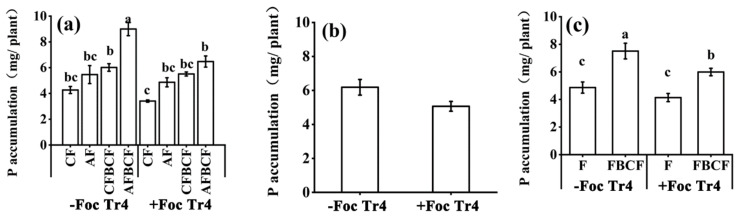
Effects of alkaline fertilizer and biocontrol fungi on P accumulation of banana 60 days after Foc Tr4 inoculation. (**a**) Effect of each treatment on P accumulation. (**b**) Effect of inoculation of pathogen (main plots) on P accumulation. (**c**) Effect of biocontrol fungi (sub-plots) on P accumulation. Different letters on the top of the bars indicated significant difference (*p* < 0.05) among the treatments. CF stands for conventional fertilizer. AF stands for alkaline fertilizer. CFBCF stands for conventional fertilizer and biocontrol fungi. AFBCF stands for alkaline fertilizer and biocontrol fungi.

**Figure 7 jof-08-00261-f007:**
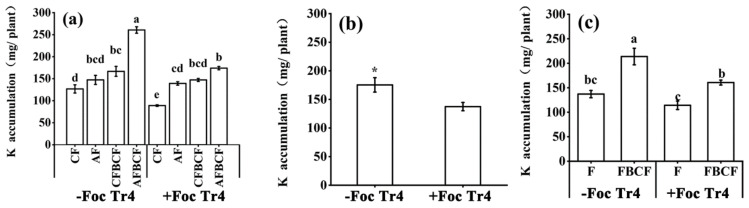
Effects of alkaline fertilizer and biocontrol fungi on K accumulation in banana 60 days after Foc Tr4 inoculation. (**a**) Effect of each treatment on K accumulation. (**b**) Effect of inoculation of pathogen (main plot) on K accumulation. (**c**) Effect of biocontrol fungi (sub-plots) on K accumulation. Different letters on top of the bars indicated significant difference (*p* < 0.05) among the treatments. * indicated that there was significant difference between −Foc Tr4 and +Foc Tr4 (0.01< *p* ≤ 0.05). CF stands for conventional fertilizer. AF stands for alkaline fertilizer. CFBCF stands for conventional fertilizer and biocontrol fungi. AFBCF stands for alkaline fertilizer and biocontrol fungi.

**Figure 8 jof-08-00261-f008:**
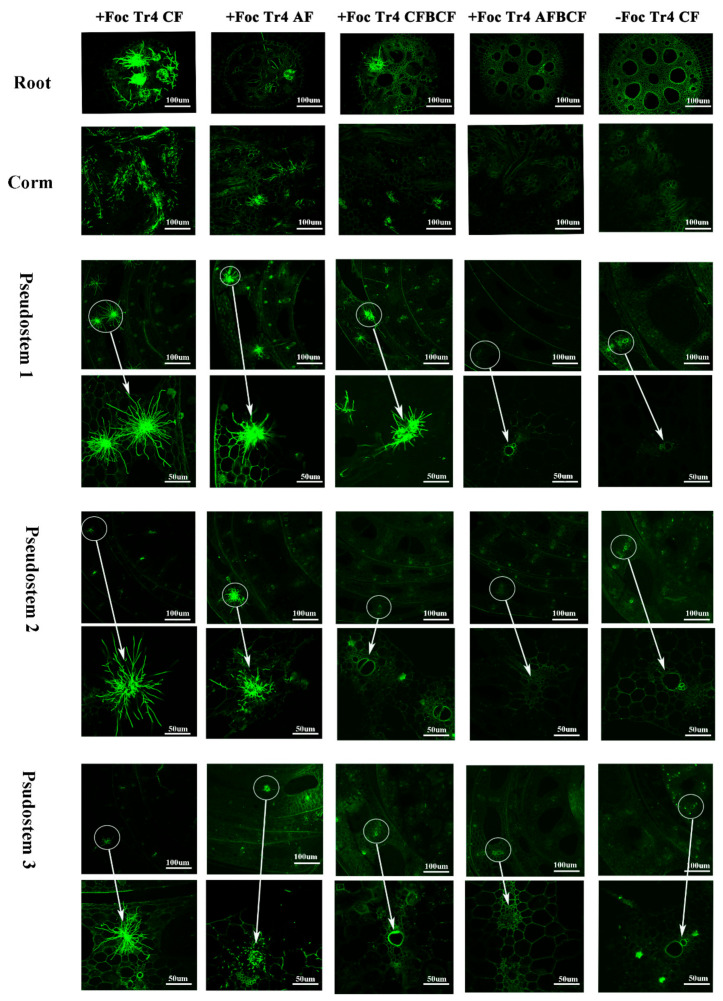
Expansion and colonization of GFP-tagged Foc Tr4 pathogen in banana in roots, rhizomes and pseudostem. Roots, rhizomes, and pseudostems were observed in cross sections. Pseudostems 1, 2, and 3 represented the pseudostem from rhizome to the first leaf, the first leaf to the second leaf and the second leaf to the third leaf count backwards, respectively. The electron microscope image indicated by the arrow is a fivefold magnification of the pathogen in the circle; The typical fluorescence image of Foc Tr4 pathogen in bananas is shown in the left column of Figure 8. The obvious hyphae colony image was the Foc Tr4 pathogen and the others were the plants’ own fluorescence. CF stands for conventional fertilizer. AF stands for alkaline fertilizer. CFBCF stands for conventional fertilizer and biocontrol fungi. AFBCF stands for alkaline fertilizer and biocontrol fungi.

**Figure 9 jof-08-00261-f009:**
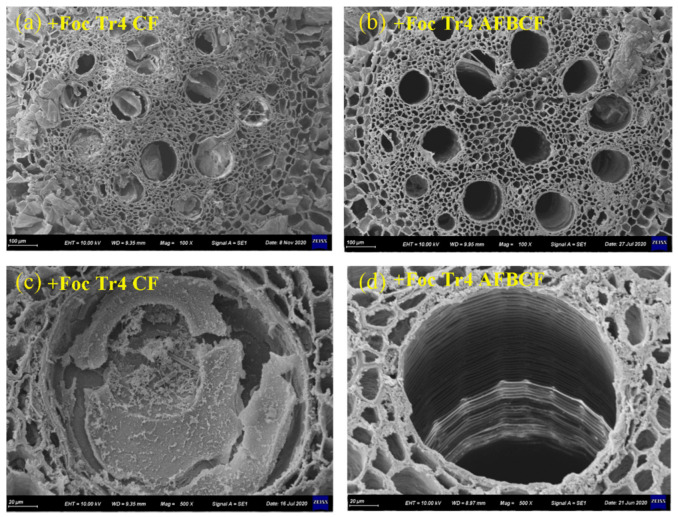
SEM images of banana root vessels in +Tr4 CF and +Tr4 AFBCF treatment. (**a**) Most of the vessels for +Tr4 CF treatment of roots were blocked by tylosis; (**b**) The vessels of +Tr4 AFBCF root were hardly found to be blocked by tylosis; (**c**) +Tr4 CF vessels were blocked by tyloses; (**d**) +Tr4 AFBCF vessels were unobstructed. CF stands for conventional fertilizer. AFBCF stands for alkaline fertilizer and biocontrol fungi.

**Table 1 jof-08-00261-t001:** Browning section area of banana rhizome 60 days after Foc Tr4 inoculation.

Treatment		Total Area (cm^2^)	Browning Area (cm^2^)	Proportion of Browning Area (%)
−Foc Tr4	CF	2.41 ± 0.15 d	——	——
	AF	3.45 ± 0.19 c		
	CFBCF	4.81 ± 0.04 b		
	AFBCF	5.73 ± 0.37 a		
+Foc Tr4	CF	1.78 ± 0.06 e	1.39 ± 0.05 a	77.86 ± 3.75 a
	AF	1.81 ± 0.04 e	0.99 ± 0.03 b	54.90 ± 0.77 b
	CFBCF	1.90 ± 0.08 d,e	0.56 ± 0.05 c	29.53 ± 2.81 c
	AFBCF	2.99 ± 0.18 c	0.46 ± 0.04 c	15.47 ± 1.42 d

Date in the table were average ± SE. Different small case letters in the same column indicate significant differences among treatments according to One-Way ANOVA tests (*p* < 0.05). CF stands for conventional fertilizer. AF stands for alkaline fertilizer. CFBCF stands for conventional fertilizer and biocontrol fungi. AFBCF stands for alkaline fertilizer and biocontrol fungi.

**Table 2 jof-08-00261-t002:** Effect of alkaline fertilizer and biocontrol fungi on banana height and stem diameter increments, shoot and root dry weight 60 days after Foc Tr4 inoculation.

Treatment		Height Increments (cm)	Average 1	Average 2
−Foc Tr4	CF	8.16 ± 0.29 b	8.47 ± 0.30 b	9.04 ± 0.32 **
	AF	8.78 ± 0.52 b		
	CFBCF	8.58 ± 0.27 b	9.61 ± 0.52 a	
	AFBCF	10.64 ± 0.79 a		
+Foc Tr4	CF	5.72 ± 0.37 c	5.87 ± 0.22 c	6.74 ± 0.28
	AF	6.02 ± 0.24 b,c		
	CFBCF	6.74 ± 0.25 c	7.60 ± 0.33 b	
	AFBCF	8.46 ± 0.25 b		
Treatment		Stem diameter increments (mm)	Average 1	Average 2
−Foc Tr4	CF	4.28 ± 0.44 c,d	4.39 ± 0.41 b	5.16 ± 0.36 *
	AF	4.50 ± 0.75 c,d		
	CFBCF	4.96 ± 0.60 b,c	5.93 ± 0.48 a	
	AFBCF	6.90 ± 0.45 a		
+Foc Tr4	CF	2.46 ± 0.18 e	2.83 ± 0.32 c	4.11 ± 0.40
	AF	3.20 ± 0.61 b,d,e		
	CFBCF	4.42 ± 0.28 c,d	5.39 ± 0.45 a,b	
	AFBCF	6.36 ± 0.60 a,b		
Treatment		Shoot dry weight (g/plant)	Average 1	Average 2
−Foc Tr4	CF	1.62 ± 0.11 c	1.75 ± 0.12 b,c	2.02 ± 0.11 *
	AF	1.89 ± 0.20 b,c		
	CFBCF	1.92 ± 0.12 b,c	2.29 ± 0.14 a	
	AFBCF	2.65 ± 0.05 a		
+Foc Tr4	CF	1.30 ± 0.02 d	1.58 ± 0.10 c	1.75 ± 0.07
	AF	1.85 ± 0.07 b,c		
	CFBCF	1.79 ± 0.05 b,c	1.93 ± 0.06 b	
	AFBCF	2.06 ± 0.05 b		
Treatment		Root dry weight (g/plant)	Average 1	Average 2
−Foc Tr4	CF	0.20 ± 0.03 b	0.21 ± 0.02 b	0.25 ± 0.02 *
	AF	0.23 ± 0.02 b		
	CFBCF	0.19 ± 0.01 b,c	0.29 ± 0.04 a	
	AFBCF	0.39 ± 0.02 a		
+Foc Tr4	CF	0.14 ± 0.01 c	0.15 ± 0.01 c	0.18 ± 0.01
	AF	0.17 ± 0.01 b,c		
	CFBCF	0.20 ± 0.01 b	0.21 ± 0.01 b	
	AFBCF	0.21 ± 0.01 b		

Data in the table were average ± SE. Different small case letters in the same column indicate significant difference (*p* < 0.05) among the treatments. * indicated that there was significant difference between with and without inoculation of Foc Tr4 (0.01 < *p* ≤ 0.05). ** indicated that there was extremely significant difference between with and without inoculation of the Foc Tr4 (*p* ≤ 0.01). CF stands for conventional fertilizer. AF stands for alkaline fertilizer. CFBCF stands for conventional fertilizer and biocontrol fungi. AFBCF stands for alkaline fertilizer and biocontrol fungi.

**Table 3 jof-08-00261-t003:** Effect of alkaline fertilizer and biocontrol fungi on antioxidant enzyme activities of banana root.

Treatment		POD Activity (U/g. min)	Average 1	Average 2
−Foc Tr4	CF	24.17 ± 1.27 e	23.47 ± 0.82 d	26.32 ± 0.97
	AF	22.78 ± 1.11 e		
	CFBCF	28.89 ± 0.28 d	29.17 ± 0.48 c	
	AFBCF	29.44 ± 1.00 d		
+Foc Tr4	CF	34.44 ± 0.73 c	36.11 ± 0.82 b	38.54 ± 0.88 **
	AF	37.78 ± 0.28 b		
	CFBCF	40.83 ± 0.83 a	40.97 ± 0.62 a	
	AFBCF	41.11 ± 1.11 a		
Treatment		CAT activity (mg/g. min)	Average 1	Average 2
−Foc Tr4	CF	6.39 ± 0.26 b,c	6.51 ± 0.17 c	5.88 ± 0.28
	AF	6.62 ± 0.25 b,c		
	CFBCF	5.49 ± 0.63 c	5.26 ± 0.40 d	
	AFBCF	5.03 ± 0.60 c		
+Foc Tr4	CF	7.58 ± 0.39 b	8.40 ± 0.41 b	9.38 ± 0.42 **
	AF	9.23 ± 0.06 a		
	CFBCF	10.36 ± 0.98 a	10.36 ± 0.48 a	
	AFBCF	10.36 ± 0.41 a		
Treatment		SOD activity (mg/g. min)	Average 1	Average 2
−Foc Tr4	CF	280.11 ± 7.84 c	278.43 ± 4.86 c	283.19 ± 3.70
	AF	276.75 ± 7.35 c		
	CFBCF	285.71 ± 7.00 c	287.96 ± 5.26 c	
	AFBCF	290.20 ± 9.17 c		
+Foc Tr4	CF	338.38 ± 2.96 b	343.42 ± 2.66 b	350.98 ± 2.82 **
	AF	348.46 ± 1.12 a,b		
	CFBCF	354.06 ± 1.12 a,b	358.54 ± 2.24 a	
	AFBCF	363.03 ± 1.94 a		

Different letters in the small case same column indicated significant difference (*p* < 0.05) among the treatments. ** indicated that there was extremely significant difference between with and without inoculation of the Foc Tr4 (*p* < 0.01). POD, CAT and SOD in the table represent peroxidase, catalase and superoxide dismutase, respectively. CF stands for conventional fertilizer. AF stands for alkaline fertilizer. CFBCF stands for conventional fertilizer and biocontrol fungi. AFBCF stands for alkaline fertilizer and biocontrol fungi.

## Data Availability

The raw data supporting the conclusions of this manuscript will be made available by the authors, without undue reservation, to any qualified researcher.

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
