# Peer review of "Synchronized Efficacy and Mechanism of Alkaline Fertilizer and Biocontrol Fungi for Fusariumoxysporum f. sp. cubense Tropical Race 4"

_jof, 2022, doi:10.3390/jof8030261_

Round 1

Reviewer 1 Report

The paper describes an extensive study on synergistic effect of biocontrol fungi and alternative fertilization on the severity of fusarium wilt in banana plants caused by Fusarium oxysporum artificial infection. The manuscript is reasonably well-written and presents new data, however, it should be carefully revised before it can be processed further. Some specific comments and suggestions were given below.

Abstract should be re-written. It contains a bunch of methodology details, which are unnecessary and distract the reader from the main topic of the study. Try to stay focus on the most important things.

Introduction is too long. The section should be shortened by at least half a page. It should also be less chaotic. Try to develop the logical order of topics introduced and do not jump between them. I understand that there is a lot to signalize but I believe that the present version can be improved.

Materials and Methods are exhaustive and this description allows to repeat the experiments. Paragraph in lines 146-151 could be more clear, explaining the percentage given for compounds in the brackets. Also, sections 2.4.5. and 2.4.7. could be expanded to give more details or, alternatively, references to previous works.

Results and Discussion should definitely be separated, since there is a lot of results presented and very little of discussion. The following paragraphs should be moved to Discussion for a start: lines 326-337, lines 366-379, lines 441-456, lines 562-574 and lines 582-593. This is of course not enough and the discussion should be expanded substantially, as some of the results were not discussed at all.

Author Response

Response to reviewer 

1 Abstract should be re-written. It contains a bunch of methodology details, which are unnecessary and distract the reader from the main topic of the study. Try to stay focus on the most important things.

The abstract has been rewritten as suggested and unnecessary details have been canceled. All of the changes are presented in 'Track Changes' mode which you will find in the manuscript.

2 Introduction is too long. The section should be shortened by at least half a page. It should also be less chaotic. Try to develop the logical order of topics introduced and do not jump between them. I understand that there is a lot to signalize but I believe that the present version can be improved.

Introduction has been shortened as suggested by more than half page.

3 Materials and Methods are exhaustive and this description allows to repeat the experiments. Paragraph in lines 146-151 could be more clear, explaining the percentage given for compounds in the brackets. Also, sections 2.4.5. and 2.4.7. could be expanded to give more details or, alternatively, references to previous works.

The authors of this paper add explanations and references in the corresponding sections according to the reviewers’ suggestion as follows.

1) Paragraph in lines 146-151 could be more clear, explaining the percentage given for compounds in the brackets.

The fertilizer was prepared by adding urea (N content accounted for 46%), potassium chloride (K2O content accounted for 60%) and sodium dihydrogen phosphate dihydrate (P2O5 content accounted for 45%) into urea-formaldehyde solution during the urea-formaldehyde formation. N: P2O5: K2O ratio of the liquid fertilizer was 2: 0.5: 1.

2) sections 2.4.5. and 2.4.7. could be expanded to give more details or, alternatively, references to previous works.

The above dried roots, stems and leaves powder sample (passed to 0.25mm sieve) was digested by hydrogen peroxide (H2O2)-sulfuric acid (H2SO4) [36].

The SEM samples were prepared according to NebesáÅ™ová's method [38].

4 Results and Discussion should definitely be separated, since there is a lot of results presented and very little of discussion. The following paragraphs should be moved to Discussion for a start: lines 326-337, lines 366-379, lines 441-456, lines 562-574 and lines 582-593. This is of course not enough and the discussion should be expanded substantially, as some of the results were not discussed at all.

The authors have separated the results analysis from discussion based on the review’s recommendations. All the results are discussed in detail and are presented in the manuscript in 'Track Changes' mode.

Reviewer 2 Report

The manuscript by Yuanqiong Li et alii is devoted to control of an important plant pathogen via a special fertilization management as well as by adding three different well-known fungi. The aim of this study is important and worth to be followed.

The procedure is tested with the agriculturally important banana wilt, induced by a special variety of Fusarium oxysporum forma cubense

Thewre are, however, many important drawbacks that prevent publication at the present stage. 

Title: Taxonomy first: Despite of the other words in capitals, species names etc must be given as prescribed by the rules. 
I recommend to avoid the term "synchronizes", as it should be used for events happening in the same time courses. Do the authors mean 'coordinated'? 

The authors are adviced to use a more focussed use of language. An example is seen already in the Highlights section. "There existed ... of" gives away the chance to use the most important subject of the sentence also as grammatical subject.  A possible improvement could be: 'Banana Fusarium wilt is controlled by combined application of alkaline fertilizer and the biocontrol fungi ..." Could the authors please provide the species name of the Trichoderma isolate used?

The authors should avoid unnecessary abbreviations. There is no need for "CF" (only an example; there is more), especially, but not exclusively in the Highlights and Abstract sections. 

Observations must be given in present tense. Consequently, in Highlight 1 and elsewhere "existed" must be 'exist', in case the authors stay with this unnecessary term. 

Highlight 2: Probably the authors want to point out: 'Synergistic application of alkaline fertilizer and biocontrol fungi improves banana root activity (no abbreviation recommended, by the way) ...'

Highlight 3: Again: The authors can easily express there results much clearer. The reviewer recommends: 'The application of alkaline fertilizer and biocontrol fungi reduces root colonization, spreading from roots to rhizome ..., and prevents tylosis formation in vascular vessels.'

These are only examples; the complete text must be improved with respect to clear wording. 

Abstract: Again, the authors are adviced not to give away main statements to subordinate clauses. Main things should be given in main clauses. The reviewer recommends: 'Effects and mechanism of ... determined in this study', or similar. 
Also: You cannot combine a fertilzer "without biocontrol fungi". Please revise the complete text for language logic and clarity. 
In addition: Revise the complete text for correct grammar: the Abstract example: "synchronizing use of ..." must read 'synchronized use of...'. Again: Avoid to place main things in subordinate clauses. "The results showed that ..." Better: 'Synchronized use of ...  eliminates ... and reduces ...'

Please give only reasonable and statistically relevant numbers of digits. "64.88% and 65.43%" are beyond everything that can reasonably be measured in such tests. Please check the complete text in this respect.

Please avoid abbreviations especially in the Abstract section. These texts are available in databases and must be readable without any further reference. 

Materials and Methods: The authors are adviced to avoid unusual writing. There is no need for "~". Like everywhere the text must be revised very carefully also for seemingly small things.

"... plant height and stem diameter the plantlets ..." Is obviously not a correct sentence. The authors should keep in mind that too many seemingly less important mistakes distract readers considerably. At this point - latest - the reviewer would normally not have read the paper further. Careful revision increases impact of a publication considerably. 

Please provide exact recipes for the fertilizers employed; compound and amount must be given accurately. Materials and Methods is a very important part of a publication, because absolute accuracy is the only way for warranting reproducibility. The use of chemical formulae is beyond any logic. Examples only: K20 is certainly not potassium chloride; sodium dihydrogen phosphate is certainly not P2O5; the percentages given in parantheses are not at all self-explicative. The reviewer sees what the authors probably want to say. Why not make things clear?

Avoid lab slang expressions. There is nothing like "Milli-Q water". 
Please provide either recipes or the original citations for media and procedures used. 

Results and Discussion
Especially  this section is extremely hard to read. This is mainly due to mixing of "Results" and "Discussion". This is not approriate for this type of publication. The reviewer insists on clearly separating "Results" from "Discussion".

Figures and Tables  must be understandable without constant referring to the main text. Thus, more detailed, accurate figure caption are mandatory. If abbreviations are used, they need to be explained. 
What do the inscripts a - e in Table 1, 2 exactly mean? 

On the whole, the reviewer gains the impression thta the observations and data provided by the authors are sound. Unfortunately, the manuscript is extremely hard to read due to many drawbacks in structuring and writing the manuscript. The reviewer has sincere doubts that it will be read by more than a few scientists. In its present form, the reviewer does not recommend this manuscript for publication. It is mandatory to revise all parts very carefully. 

Author Response

1 Title: Taxonomy first: Despite of the other words in capitals, species names etc must be given as prescribed by the rules.

I recommend to avoid the term "synchronizes", as it should be used for events happening in the same time courses. Do the authors mean 'coordinated'?

1)The title has been revised as follows according your suggestion. Corresponding parts of the manuscript have also been revised.

Synchronized Efficacy and Mechanism of Alkaline Fertilizer and Biocontrol Fungi for Fusarium oxysporum f. sp. cubense Tropical Race 4

2)In the experiment, the alkaline fertilizer and the biocontrol fungi were applied simultaneously. The "synchronized use of " was exactly to express the fertilization and application of biocontrol fungi in the same time.

2 The authors are adviced to use a more focussed use of language. An example is seen already in the Highlights section. "There existed ... of" gives away the chance to use the most important subject of the sentence also as grammatical subject.  A possible improvement could be: 'Banana Fusarium wilt is controlled by combined application of alkaline fertilizer and the biocontrol fungi ..." Could the authors please provide the species name of the Trichoderma isolate used?

  1)Modifications have been made according to recommendations. The revised highlights are as follows.

Banana Fusarium wilt is controlled by combined application of alkaline fertilizer and the biocontrol fungi which consisted of non-pathogenic Fusarium oxysporum, Trichoderma harzianum QL18-8 and Paecilomyces lilacinus.

  2)Trichoderma harzianum QL18-8 was the Trichoderma used in the study. Trichoderma in original manuscript has been replaced by Trichoderma harzianum QL18-8. Trichoderma harzianum Ql18-8 is isolated from Qinling, China in 2018.

3 The authors should avoid unnecessary abbreviations. There is no need for "CF" (only an example; there is more), especially, but not exclusively in the Highlights and Abstract sections.

Unnecessary abbreviations in the manuscript have been deleted. The CF in the manuscript is not an abbreviation. It is a code for treatment of conventional fertilizer.

4 Observations must be given in present tense. Consequently, in Highlight 1 and elsewhere "existed" must be 'exist', in case the authors stay with this unnecessary term.

The manuscript has been revised according to suggestions as follows.

Highlight 1: Banana Fusarium wilt is controlled by combined application of alkaline fertilizer and the biocontrol fungi which is composed of non-pathogenic Fusarium oxysporum, Trichoderma and Paecilomyces lilacinus.

5 Highlight 2: Probably the authors want to point out: 'Synergistic application of alkaline fertilizer and biocontrol fungi improves banana root activity (no abbreviation recommended, by the way) ...'

Modifications have been made according to recommendation as follows.

Highlight 2: Synergistic application of alkaline fertilizer and biocontrol fungi improves banana root activity, induces banana defense responses, promotes banana NPK accumulation.

6 Highlight 3: Again: The authors can easily express there results much clearer. The reviewer recommends: 'The application of alkaline fertilizer and biocontrol fungi reduces root colonization, spreading from roots to rhizome ..., and prevents tylosis formation in vascular vessels.' These are only examples; the complete text must be improved with respect to clear wording.

The expression in the manuscript has been checked and revised as suggested.

Highlight 3: The application of alkaline fertilizer and biocontrol fungi reduces the pathogen colonization in banana roots, spreading from root to rhizome and pseudostem, and prevents tylosis formation in vascular vessels.

7 Abstract: Again, the authors are adviced not to give away main statements to subordinate clauses. Main things should be given in main clauses. The reviewer recommends: 'Effects and mechanism of ... determined in this study', or similar.

Also: You cannot combine a fertilzer "without biocontrol fungi". Please revise the complete text for language logic and clarity. In addition: Revise the complete text for correct grammar: the Abstract example: "synchronizing use of ..." must read 'synchronized use of...'. Again: Avoid to place main things in subordinate clauses. "The results showed that ..." Better: 'Synchronized use of ...  eliminates ... and reduces ...'

Thank you for your suggestions. The authors have revised and rewritten the abstract according to the suggestion. All of the changes are presented in 'Track Changes' mode which you will find in the manuscript.

8 Please give only reasonable and statistically relevant numbers of digits. "64.88% and 65.43%" are beyond everything that can reasonably be measured in such tests. Please check the complete text in this respect.

The relevant numbers in the manuscript have been checked and revised as recommended. Such as 64.88% and 65.43% have been revised into 65%, and 65% and 31.17% and 33.33% into 31% and 33% respectively.

9 Please avoid abbreviations especially in the Abstract section. These texts are available in databases and must be readable without any further reference.

The unnecessary abbreviations in the whole manuscript have been canceled as reviewer suggested. There is no abbreviation in abstract. However, several abbreviations are retained in the manuscript for concise expression.

10 Materials and Methods: The authors are adviced to avoid unusual writing. There is no need for "~". Like everywhere the text must be revised very carefully also for seemingly small things.

Thank you for your suggestion. The "~" in the manuscript has been corrected.

11 "... plant height and stem diameter the plantlets ..." Is obviously not a correct sentence. The authors should keep in mind that too many seemingly less important mistakes distract readers considerably. At this point - latest - the reviewer would normally not have read the paper further. Careful revision increases impact of a publication considerably.

The author has been corrected such mistake as you suggested. Thanks again for your kind suggestion.

12 Please provide exact recipes for the fertilizers employed; compound and amount must be given accurately. Materials and Methods is a very important part of a publication, because absolute accuracy is the only way for warranting reproducibility. The use of chemical formulae is beyond any logic. Examples only: K20 is certainly not potassium chloride; sodium dihydrogen phosphate is certainly not P2O5; the percentages given in parantheses are not at all self-explicative. The reviewer sees what the authors probably want to say. Why not make things clear?

The authors of this paper add explanations in the corresponding sections according to the reviewers’ suggestion as follows.

The fertilizer was prepared by adding urea (N content accounted for 46%), potassium chloride (K2O content accounted for 60%) and sodium dihydrogen phosphate (P2O5 content accounted for 45%) into urea-formaldehyde solution during the urea-formaldehyde formation. N: P2O5: K2O ratio of the liquid fertilizer was 2: 0.5: 1.

13 Avoid lab slang expressions. There is nothing like "Milli-Q water". Please provide either recipes or the original citations for media and procedures used.

The author has been corrected the "Milli-Q water" into "deionized water".

Results and Discussion

14 Especially this section is extremely hard to read. This is mainly due to mixing of "Results" and "Discussion". This is not approriate for this type of publication. The reviewer insists on clearly separating "Results" from "Discussion".

The authors have separated the results analysis from discussion based on the review’s recommendations.

15 Figures and Tables must be understandable without constant referring to the main text. Thus, more detailed, accurate figure caption are mandatory. If abbreviations are used, they need to be explained. What do the inscripts a - e in Table 1, 2 exactly mean?

  1) Notes about abbreviations in figures and tables have been made and necessary full names have been added as suggested.

  2) The inscripts a - e in Table 1, 2 are normal expression to indicates the difference level according to statistics.

In table 1: The authors have revised the “Different letters in the same column indicate significant differences among treatments according to One-Way ANOVA tests (P < 0.05)” in to “Different small case letters in the same column indicate significant differences among treatments according to One-Way ANOVA tests (P < 0.05)” in table 1,

In table 2: The authors have revised the “Different letters in the same column indicate significant difference (P<0.05) among the treatments” in to “Different small case letters in the same column indicate significant difference (P<0.05) among the treatments” in table 2.

Reviewer 3 Report

I can confirm that the subject matter of this research article is of interest and relevance for publication in Journal of Fungi. The paper is well structured and includes original data that maybe of interest for the readers. Abstract – the information given is adequate and concise. Introduction, literature review sections of paper is adequately exploited, well documented by figures and tables.

The manuscript is well written, but I have only minor comments to the Authors:

-add reference to Data Analysis

-in my opinion Conclusion may be improved giving few key message/take home message to the readers. An idea may be to synthetize in 3-5 bullet the key results of evidences and recommendation. This improvement will increase clearness and readability. Add a practical implications statement

Author Response

1 Add reference to Data Analysis

The results have been described separately from the discussion in the revised manuscript and corresponding references have been added to the discussion.

2 In my opinion Conclusion may be improved giving few key message/take home message to the readers. An idea may be to synthetize in 3-5 bullet the key results of evidences and recommendation. This improvement will increase clearness and readability. Add a practical implications statement

The conclusions have been divided into the following three parts according to the recommendations.

(1) The combination application of alkaline fertilizer and biocontrol fungus could significantly reduce IYL, rhizome browning degree, and banana Fusarium wilt incidence.

(2) The synergistic effects of alkaline fertilizer and biocontrol fungus significantly increased root activity and root antioxidant enzyme activity of banana plants. Meanwhile, the absorption and accumulation of nitrogen, phosphorus and potassium nutrients in banana plants were boosted, so as to improve the growth of banana and enhance the plant to resist the Fusarium wilt disease.

(3) The mechanism of banana Fusarium wilt control by application of the alkaline fertilizer and biocontrol fungus was that their synergistic effect would significantly reduce the Foc Tr4 pathogen infection and colonization in banana root and subsequently inhabit the pathogen expanding to rhizome and pseudostem. Tylosis formation was then prevented and thus the vascular vessel blockage was avoided to ensure the normal transport of water and nutrients between underground parts to aboveground ones.

Round 2

Reviewer 1 Report

The revised manuscript was greatly improved. All comments and suggestions of the reviewer were addressed and properly incorpoorated in the revision.

Reviewer 2 Report

The manuscript has been improved considerably and can be accepted for publication. 

This manuscript is a resubmission of an earlier submission. The following is a list of the peer review reports and author responses from that submission.

Round 1

Reviewer 1 Report

I have added notes with edits, feedback and comments, to the attached PDF version of the paper.

On one hand, this paper presents very interesting results that certainly merit publication.

On the other hand, the paper needs a complete overhaul [by e.g. an English editor]. Nearly each and every sentence needs to be rewritten. The paper cannot be published in its current form. 

Reviewer 2 Report

Fusarium wilt is one of the main banana diseases, and the tropical race 4 (TR4) is the principal variant threatening the crop. The lack of methods to effectively manage the disease impose many difficulties in worldwide banana production. The manuscript studied the effects of alkaline fertilizer combined with three biological control agents. The studies were conducted in greenhouses experiments on which the disease, phenological and physiological parameters, and fluorescent microscopy were used to understand the differences among treatments. The investigation has scientific merit and was well designed. Despite this, I noticed the lack of one control which is essential for some of the interpretations. However, my main concern is: with the lack of scientific writing, use of jargon, the excessive use of word conjunctions; 25% of the literature cited are unavailable for the international community, including some methods that are not reproducible due to the literature; the lack of statistical support for interpretations of most of the results, and the lack of description of the statistical methods used. Most of the discussion and key points of the study were made without appropriate statistical support or were not observed and tested on this study. Additionally, I strongly encourage the authors to check the guidelines for scientific writing and send them for review to additional colleagues before submitting the manuscript. 

Specific and overall comments in the Introduction:

  • An extensive review of English is needed. I strongly recommend the authors send the manuscript to an English speaker (native or fluent) colleague or use an English professional translator before submitting any manuscript. There are many incorrect sentences in the introduction section (mainly) and the subsequent paragraphs.
  • Avoid the use of conjunction words, i.e., therefore, on the other hand, however, etc. 
  • All scientific names MUST be italicized.
  • The word "Fusarium" in Fusarium wilt is a proper name, and the first letter MUST be capitalized. 
  • The authors mixed Material and Methods (MM) in the last paragraph of the Introduction.
  • Many parts of the Introduction contain questionable sentences and affirmations whose references are unavailable for the international community.
  • The authors did not cite many vital references in this pathosystem. 

The MM section must improve. See some key points:

  • Many of the references are not available for the international community. For these methods, I strongly recommend the authors describe them (TTC, POD, CAT, SOD, and He's method).
  • Some results are not described in MM. For example, the browning area (Table 1).
  • I am unsure if the authors used the Tropical Race 4 (TR4) or Subtropical Race 4 (SR4) for the study. 
  • Statistical tests were not described in MM.
  • There is a formula in MM which is not in English.
  • There is a lack of biological controls (BCF control). For example, in the Results section (lines 367-369), the single BCF treatment is not shown in table 2. 

Comments in the Results and Discussion section:

  • Most of the results, discussions, and comments do not have statistical support. The authors affirmed that there were differences between treatments without performing statistical analysis. I strongly recommend authors add the exact P-values in the following examples: Lines 270-272; 281-283; 283-284; etc.
  •  I strongly discourage authors from using the words "obvious", "obviously," or "clearly”. This is mainly due to the lack of statistical analysis to support the “obvious” findings.
  • The sentence in lines 330-332 and 337-339 are false. There is no support from this study for the affirmations. 
  • Figure 3: Regression analyses were not appropriately applied for the average of treatments. I would suggest applying it individually for each treatment. Legends are illegible.
  • At present, the authors did not conduct the statistical analyses appropriately. The analysis of variance was not performed for the results, and the multiple comparison test was not informed.
  • There is a lack of scientific writing in the whole document.    
  • Instead of using the overall P-value (< 0.05), I strongly recommend the authors use the exact P-value of the ANOVA, GLM, multiple comparison test, etc. 
  • Most of the sentences in the discussion are assumptions without support from the literature. 
  • As an overall tip, frequently observed in the whole document, it is not recommended to start sentences with code, i.e., line 399.
  • Figures 4 to 7 have three graphs each (a, b, and c). Is the same data presented in a different format? It is not appropriate. When performing statistical analyses, first of all, look for the test’s assumptions, followed by variance analyses. For the variance analyses, look for interactions first. If significant, no need to look for differences among individual factors later. In the end, a multiple comparison test can be conducted to see for differences among treatments.
  • Different letters sizes were used in the text. Example: lines 447 x 448.
  • Banana body? What do the authors mean by that? 
  • Line 577: What is this “number of pathogens”? How many pathogens are in the study? Also, if you have these “numbers”, did they count them? If so, it is strongly recommended to present them! By my understanding, this is not possible by fluorescence microscopy, mainly with the pictures shown in this study. This is misinformation. 
  • The frequently mentioned synergistic effect was not measured due to the lack of BCF control.